# The Therapeutic Role of SGLT-2 Inhibitors in Acute Heart Failure: From Pathophysiologic Mechanisms to Clinical Evidence with Pooled Analysis of Relevant Studies across Safety and Efficacy Endpoints of Interest

**DOI:** 10.3390/life12122062

**Published:** 2022-12-08

**Authors:** Dimitrios Patoulias, Nikolaos Fragakis, Manfredi Rizzo

**Affiliations:** 1Second Department of Internal Medicine, European Interbalkan Medical Center, 57001 Thessaloniki, Greece; 2Outpatient Department of Cardiometabolic Medicine, Second Department of Cardiology, General Hospital “Hippokration”, Aristotle University of Thessaloniki, 54642 Thessaloniki, Greece; 3Promise Department, School of Medicine, University of Palermo, 90127 Palermo, Italy

**Keywords:** SGLT-2 inhibitors, acute heart failure, cardiovascular disease, type 2 diabetes mellitus, mechanism, outcome

## Abstract

(1) Background: Sodium-glucose co-transporter-2 (SGLT-2) inhibitors constitute a novel drug class with remarkable cardiovascular benefits for patients with chronic heart failure (HF). Recently, this class has been utilized in acute HF as an additional treatment option to classic diuretics, which remain the cornerstone of treatment. (2) Methods: We attempted to identify those pathophysiologic mechanisms targeted by SGLT-2 inhibitors, which could be of benefit to patients with acute HF. We then conducted a comprehensive review of the literature within the PubMed database in order to identify relevant studies, both randomized controlled trials (RCTs) and observational studies, assessing the safety and efficacy of SGLT-2 inhibitors in acute HF. (3) Results: SGLT-2 inhibitors induce significant osmotic diuresis and natriuresis, decrease interstitial fluid volume and blood pressure, improve left ventricular (LV) function, ameliorate LV remodeling and prevent atrial arrhythmia occurrence, mechanisms that seem to be beneficial in acute HF. However, currently available studies, including six RCTs and two real-world studies, provide conflicting results concerning the true efficacy of SGLT-2 inhibitors, including “hard” surrogate endpoints. (4) Conclusions: Current evidence appears insufficient to substantiate the use of SGLT-2 inhibitors in acute HF. Further trials are required to shed more light on this issue.

## 1. The Entity of Acute Heart Failure

Acute heart failure (HF) is defined as the new onset (first manifestation of HF) or worsening of symptoms and signs of HF (acute decompensation of established, chronic HF) [1]. It is a life-threatening medical condition, requiring hospitalization and urgent therapeutic intervention [2]. It represents one of the major causes of hospitalization for elderly subjects in the United States [3]. The onset of symptoms may be gradual or rapid, modifying the intensity and severity of clinical presentation [4]. Elevated ventricular filling pressures with or without a decrease in cardiac output constitute an almost universal finding in acute HF [5]. One-month mortality after an episode of acute HF ranges from 3.1% for patients younger than 60 years old to 7% for patients older than 80 years old, while the corresponding percentages for 6-month mortality rates are 11.3% and 23.9% [6]. Various baseline characteristics were shown to have prognostic value in acute HF, with increasing age being a strong and independent predictor of worse outcomes [6,7]. Sudden cardiac death accounts for a significant proportion, equal to 17%, of all deaths observed within the first month after hospitalization for acute HF [8]. However, as demonstrated in other relevant trials, HF represents the major underlying cause of death, accounting for 38% of all recorded deaths [9]. Older subjects and those with HF with preserved left ventricular ejection fraction (HFpEF) tend to die more frequently from other causes than HF or sudden cardiac death, including non-cardiac causes as well [9].

Main pathophysiologic underlying mechanisms include left ventricular (LV) and sometimes right ventricular (RV) systolic and diastolic dysfunction, a substantial increase in preload and afterload, and sodium and water renal retention [4]. Treatment is individualized, with intravenous loop diuretics representing the cornerstone of treatment ((class I), level of recommendation C), along with oxygen and ventilatory support [4]. A continuous diuretic infusion was associated with significantly greater total urine output, body weight reduction and reduction in brain natriuretic peptide (BNP) levels compared to intermittent, bolus infusion; however, no difference in all-cause mortality and duration of hospitalization was shown [10]. However, the choice of loop diuretic regimen in acute HF is at the treating physician’s discretion [11]. The appropriate diuretic strategy needs to be further investigated, according to acute HF phenotypes [12].

Diuretic resistance, defined as “failure to decongest despite adequate and escalating doses of diuretics” appears to be a common problem among subjects hospitalized with acute HF, affecting 20–50% of all patients [13]. This problem can be overcome with the addition to the treatment regimen of a thiazide-type diuretic in order to induce diuretic synergy via sequential nephron blockade [14]. Data concerning the addition of tolvaptan, an oral vasopressin-2 receptor antagonist, to a fixed-dose furosemide regimen appear to be contradictory in the acute HF setting [15,16], with tolvaptan not being inferior to thiazide-type diuretics in terms of induced weight loss and achieved decongestion [15]. However, according to a recent network meta-analysis of relevant trials, among subjects with diuretic-resistant acute HF, no diuretic appears to be more effective when added to furosemide, compared to furosemide alone [17].

Sodium-glucose co-transporter-2 (SGLT-2) inhibitors represent a novel drug class, initially purposed for the treatment of type 2 diabetes mellitus (T2DM). They inhibit the SGLT-2 receptors predominantly expressed in the proximal tubule of the nephron, thus, they induced glycosuria, therefore osmotic diuresis and natriuresis via proximal tubular sodium loss [18]. SGLT-2 inhibitors were shown to provide significant cardiovascular benefits among subjects with established HF with reduced ejection fraction (HFrEF) by significantly decreasing the risk for cardiovascular death and hospitalization due to worsening HF [19,20], while they were also shown to significantly decrease the risk for HF decompensation among subjects with HFpEF without affecting the overall risk for cardiovascular death [21,22]. Of note, T2DM status at baseline does not affect the observed effects of SGLT-2 inhibitors in patients with established HF. Therefore, recent guidelines for the management of chronic HF recommend the use of SGLT-2 inhibitors in subjects with HFrEF, regardless of the history of T2DM [4]. The question that arises is whether this drug class might also be of benefit to patients admitted due to acute HF. In the present review article, we will discuss the available evidence regarding the clinical efficacy and safety of SGLT-2 inhibitors in acute HF.

## 2. Mechanisms Targeted by SGLT-2 Inhibitors

### 2.1. Volume Regulation

Osmotic diuresis mediated by SGLT-2 inhibitors represents a distinct mechanism of action, different from those mechanisms implicated in the action of classic diuretics. Hallow and colleagues have previously shown that dapagliflozin may be inferior to bumetanide in terms of diuresis and natriuresis; however, it is associated with a two-fold greater reduction in interstitial fluid volume compared to bumetanide, therefore, SGLT-2 inhibitors may be effective in decreasing interstitial congestion without provoking arterial underfilling [23]. Acute treatment with SGLT-2 inhibitors in subjects with preserved renal function may not have a significant effect on plasma volume and intracellular volume status; however, it correlates with a significant decrease in extracellular volume status, as seen in the DAPASALT trial [24].

Among subjects with established HFrEF, treatment with an SGLT-2 inhibitor compared to placebo resulted in a significant decrease in estimated extracellular volume and estimated plasma volume after 12 weeks [25]. In addition, early treatment with an SGLT-2 inhibitor in subjects with a recent acute myocardial infarction, pre-existing T2DM and established HF was shown to produce a significant decrease in plasma volume status after long-term treatment, equal to 24 weeks [26].

Of note, acute increase in 24 h urine volume in patients receiving SGLT-2 inhibitors was shown to significantly correlate with 24 h urinary sodium excretion but did not correlate with 24 h urinary glucose excretion [27].

Overall, SGLT-2 inhibitors induce a substantial increase in urine volume, which results in acute body weight reduction, exponentially reducing interstitial fluid volume. Long-term treatment is associated with reduced plasma volume status and reduced extracellular volume. However, it has to be admitted that SGLT-2 inhibitors increase the risk for volume depletion phenomena by 29%, an effect seen with other classes of diuretics as well [28]. 

### 2.2. Cardiac Function and Remodeling

According to a recently published, updated meta-analysis of relevant trials, SGLT-2 inhibitors were demonstrated to significantly increase LV ejection fraction (LVEF) by 2.458%, decrease LV mass (LVM) by 6.319 g, LV end-systolic volume (LVESV) by 8.44 mL and LV end-diastolic volume (LVEDV) by 9.134 mL, while they also induce a significant decrease in left atrial volume index (LAVI) by 2.791 mL/m^2^ of body surface [29]. Similar results were shown in another relevant meta-analysis, which confirmed the beneficial effect of SGLT-2 inhibition on cardiac remodeling [30]. Of note, SGLT-2 inhibitors appear to exert greater beneficial effects on LV function compared to other antidiabetic drug classes, including dipeptidyl peptidase-4 (DPP-4) inhibitors and glucagon-like peptide-1 (GLP-1) receptor agonists [31]. 

Experimental data have suggested that SGLT-2 inhibitor-mediated improvement in LV diastolic function results in lower filling pressures, which, along with induced amelioration in myocardial stiffness and fibrosis, leads to better LV performance and might to some extent explain the beneficial results of this class in HF [32]. Other mechanisms that were implicated in the observed improvement in LV function include improvement in myocardial energetics, suppression of cardiac inflammation via NLRP3 inflammasome, decrease in myocardial oxidative stress, improvement in autophagy and lysosomal degradation, decrease in ischemia-reperfusion injury and inhibition of the Na^+^/H^+^ exchange [33,34,35].

### 2.3. Blood Pressure 

SGLT-2 inhibitors were shown to induce a significant reduction in systolic blood pressure (SBP) by 2.46 mm Hg and in diastolic blood pressure (DBP) by 1.46 mm Hg among individuals with T2DM [36]. Similarly, these results were confirmed in ambulatory blood pressure monitoring, with SGLT-2 inhibitors inducing a significant decrease in 24 h SBP by 3.76 mm Hg and in 24 h DBP by 1.83 mm Hg, compared to controls [37]. 

Among subjects with HF, SGLT-2 inhibitors were recently shown to decrease SBP by 1.68 mm Hg but they did not have a significant effect on DBP levels [38].

### 2.4. Arterial Stiffness 

Increased arterial stiffness represents increased afterload, whose reduction is a major treatment target in subjects with HF, including those suffering from acute HF. It was hypothesized that SGLT-2 inhibitors ameliorate arterial stiffness, and this mechanism is one of those implicated in the observed cardio protection with this drug class [39]. However, in a recent meta-analysis of relevant randomized controlled trials, it was shown that SGLT-2 inhibitors do not affect arterial stiffness among subjects with established cardiovascular disease or at high cardiovascular risk, while they confer only a modest but significant reduction in pulse wave velocity, the “gold-standard” of arterial stiffness, among patients with co-existing T2DM [40]. Some more recently published data support the evidence that SGLT-2 inhibitors do not exert a major effect on vascular aging [41]. However, it has to be admitted that among subjects with HFrEF without baseline diabetes mellitus, empagliflozin treatment was shown to significantly decrease pulse wave velocity after 6 months of treatment compared to placebo [42]. Therefore, subjects with HF may actually experience arterial stiffness reduction with this drug class. 

### 2.5. Myocardial Energetics

SGLT-2 inhibitors were demonstrated to shift myocardial substrate utilization from glucose to other sources but they do not affect myocardial free fatty acid uptake [43]. However, other trials have documented that SGLT-2 inhibitors do not modify myocardial glucose uptake [44]. In the chronic HF setting, both experimental and human data have supported that SGLT-2 inhibitors shift myocardial fuel utilization from glucose towards the consumption of free fatty acids, ketone bodies and branched-chain amino acids, without an increase in the risk for significant ketosis, thus improving myocardial energetics, LV remodeling and function [45,46]. However, it remains unclear whether such an effect can be translated into significant clinical benefit for patients with acute HF.

### 2.6. Myocardial Flow Reserve

Similarly, data on the effect of SGLT-2 inhibitors on myocardial flow reserve remain contradictory. Administration of empagliflozin in patients with T2DM was shown not to increase myocardial flow reserve after 13 weeks of treatment [47], whereas dapagliflozin treatment among subjects with T2DM and coronary artery disease without HF was shown to induce a significant improvement in myocardial flow reserve after 4 weeks of treatment [43]. 

### 2.7. Myocardial Fibrosis

A recent meta-analysis of all relevant trials demonstrated that empagliflozin results in a significant decrease in myocardial fibrosis, inducing a significant decrease in extracellular volume, as assessed by cardiac magnetic resonance imaging [48]. Myocardial fibrosis might be predictive of adverse outcomes among subjects with recent episodes of acute HF [49]; however, amelioration of myocardial fibrosis by SGLT-2 inhibitors might not provide any significant benefit in the acute setting of HF.

### 2.8. Arrhythmic Burden

A complication of acute HF with arrhythmias was shown to have adverse prognostic implications, leading to a significantly increased risk for recurrent hospitalization and death during the first 2 months after the initial event [50]. Almost half of the recorded arrhythmias are atrial fibrillation or atrial flutter [50]. It was previously confirmed that SGLT-2 inhibitors significantly decrease the risk for atrial fibrillation/flutter by 24%, compared to control, among subjects with T2DM [51]. However, SGLT-2 inhibitors do not affect the risk for ventricular arrhythmias and sudden cardiac death, as reported in another recently published meta-analysis [52]. In the acute setting of myocardial infarction, it was recently demonstrated that prior use of SGLT-2 inhibitors among the affected subjects resulted in significantly lower rates of atrial fibrillation, ventricular tachycardia and ventricular fibrillation, while SGLT-2 inhibitor use was associated with a significant reduction in the odds for new-onset arrhythmia during hospitalization by 65% [53].

### 2.9. Hemoconcentration

Patients recently hospitalized for acute HF were shown to experience significantly lower complications or have significantly lower mortality rates after discharge, compared to patients that did not achieve sufficient hemoconcentration [54,55]. SGLT-2 inhibitor treatment was demonstrated to induce a significant increase in hematocrit and hemoglobin levels [56,57]. This can be partially attributed to a change in plasma volume status; however, SGLT-2 inhibitors also enhance erythropoiesis, via suppression of hepcidin and modulation of other iron regulatory proteins [58]. This effect can be of significant benefit to patients hospitalized with acute HF.

An overview of SGLT-2 inhibitors main effects with potential benefit in patients with acute HF is illustrated in Figure 1.

## 3. Effect of SGLT-2 Inhibitors in the Acute HF Setting on Surrogate Endpoints: Evidence from Randomized Controlled Trials and Observational Studies

### 3.1. Search Strategy and Study Selection

We searched the PubMed database from inception to 9 November 2022 for randomized controlled trials (RCTs) or observational studies enrolling hospitalized adult subjects with acute HF, assigned either to an SGLT-2 inhibitor or control (placebo or active comparator). We implemented the following search strategy: (SGLT2 inhibitor) OR (dapagliflozin) OR (canagliflozin) OR (empagliflozin) OR (sotagliflozin) OR (ertugliflozin) OR (ipragliflozin) OR (tofogliflozin) OR (bexagliflozin) OR (licogliflozin) OR (luseogliflozin) AND acute heart failure. We did not implement any filter regarding study setting, sample size or publication language. We excluded systematic reviews with or without meta-analysis, narrative reviews, case series/case reports, letters and editorials.

After a meticulous assessment of the retrieved studies at the title and abstract levels for potential inclusion in our review, two independent reviewers (D.P. and N.F.) extracted data of interest from each eligible report.

### 3.2. Crude Outcomes of Interest

We set as major endpoints of interest all-cause death and recurrent hospitalization for HF. Regarding those dichotomous variables, differences were calculated with the use of risk ratio (RR), with a 95% confidence interval (CI), after implementation of the Mantel–Haenszel (M–H) random effects formula. Statistical heterogeneity among studies was assessed by using I^2^ statistics. Analyses were performed at the 0.05 significance level, while they were undertaken with RevMan 5.4.1. software.

### 3.3. Results

Our search retrieved six RCTs [59,60,61,62,63,64] and two observational studies [65,66], in total. Of note, one trial, the Effect of Sotagliflozin on Cardiovascular Events in Patients with Type 2 Diabetes Post-Worsening Heart Failure (SOLOIST-WHF) trial [64] enrolled subjects with T2DM and a recent hospitalization for acute decompensation of HF, therefore, it did not assess the safety and efficacy of SGLT-2 inhibitors in the “strict” setting of acute HF.

### 3.4. Pooled Analysis of RCTs

Regarding all-cause mortality, SGLT-2 inhibitor treatment was shown not to affect the risk for all-cause death among subjects hospitalized with acute HF or recently decompensated HF (RR = 0.77, 95% CI; 0.59–1.01, I^2^ = 0%), as shown in Figure 2a. However, the exclusion of the SOLOIST-WHF trial in order to restrict our analysis to the acute HF setting showed that SGLT-2 inhibitor produce a significant decrease in the risk for all-cause death among individuals with acute HF, equal to 42% (RR = 0.58, 95% CI; 0.35–0.95, I^2^ = 0%), as shown in Figure 2b.

Concerning the risk for worsening HF event, SGLT-2 inhibitor treatment was shown not to affect the corresponding risk among individuals hospitalized with acute or recently decompensated HF (RR = 0.69, 95% CI; 0.43–1.10, I^2^ = 31%), as shown in Figure 3a. Of note, even in the acute HF setting, after the exclusion of the SOLOIST-WHF trial, no difference in the treatment effect was shown (RR = 0.53, 95% CI; 0.14–1.92, I^2^ = 50%), as depicted in Figure 3b.

### 3.5. Pooled Analysis of Observational Studies

Pooled analysis of relevant observational studies revealed that treatment with SGLT-2 inhibitors in subjects with acute HF resulted in a non-significant decrease in the risk for death (RR = 0.63, 95% CI; 0.25–1.61, I^2^ = 0%), as shown in Figure 4. Concerning the risk for worsening HF, SGLT-2 inhibitor treatment was demonstrated to substantially decrease the corresponding risk by 66% (RR = 0.34, 95% CI; 0.19–0.61, I^2^ = 0%), as shown in Figure 5.

### 3.6. Other Efficacy Endpoints of Interest

#### 3.6.1. Urine Output

In the EMPAG-HF trial, researchers demonstrated a significant increase in the cumulative urine output over 5 days in the empagliflozin group versus the placebo group (*p* = 0.003). In the trial performed by Tamaki and colleagues, empagliflozin was associated with a significant increase in urine output within the first 24 h of administration compared to placebo (*p* = 0.005). In the EMPA-RESPONSE-AHF trial, Danman and colleagues also demonstrated that empagliflozin treatment, compared to the placebo, resulted in a significant increase in urine output in the first and the fourth day after hospitalization (*p* = 0.013 and 0.02, respectively). In the real-world setting, Perez-Belmonte et al. showed that empagliflozin was associated with a significant increase in urine output at discharge compared to the basal–bolus insulin group (*p* = 0.037).

#### 3.6.2. Body Weight

No significant difference in body weight reduction was shown with empagliflozin versus placebo in the EMPAG-HF trial (*p* = 0.198). In the trial performed by Charaya and colleagues, dapagliflozin treatment was associated with a more pronounced weight loss during hospitalization compared to the placebo (*p* = 0.02). In the trial conducted by Tamaki and colleagues, no significant difference in the change in body weight achieved with empagliflozin versus placebo at day 7 was documented (*p* = 0.205). Similarly, in the trial performed by Danman et al., no significant difference between achieved weight loss between empagliflozin and the placebo at day 4 was shown. Rest trials and observational studies did not report data concerning body weight change with SGLT-2 inhibitors versus control.

#### 3.6.3. N-Terminal Pro-Brain Natriuretic Peptide (NT-proBNP) Levels

In the EMPAG-AHF trial, researchers demonstrated a significantly greater reduction in NT-proBNP levels with empagliflozin compared to placebo at day 5 of hospitalization (*p* < 0.001). Similarly, in the EMPULSE trial, researchers documented that empagliflozin compared to placebo resulted in a significantly greater change in NT-proBNP levels at day 30 (*p* < 0.05). Tamaki and colleagues also documented a similar trend with empagliflozin versus placebo in terms of NT-proBNP reduction at day 7 (*p* = 0.04). However, in the EMPA-RESPONSE-AHF trial, no difference in the change of NT-proBNP levels between empagliflozin and placebo was shown (*p* = 0.63).

In the real-world setting, Perez-Belmonte et al. documented that empagliflozin compared to a basal–bolus insulin regimen resulted in a significant reduction in NT-proBNP levels at discharge (*p* = 0.021).

#### 3.6.4. Renal Function

No significant difference in the change in estimated glomerular filtration rate (eGFR) between empagliflozin and placebo was shown in the EMPAG-HF trial at different time points after randomization. In the trial by Charaya and colleagues, dapagliflozin was associated with a more pronounced decrease during hospitalization compared to the placebo (*p* = 0.049). No significant change in creatinine clearance between empagliflozin and placebo was demonstrated in the EMPULSE trial. No statistical difference in worsening renal function events, defined as an increase in serum creatinine by ≥0.3 mg/dL above baseline within 7 days of randomization, between the empagliflozin and the placebo groups was shown in the trial performed by Tamami and colleagues. Similar results were presented by trialists in the EMPA-RESPONSE-AHF trial. In the SOLOIST-WHF trial, researchers showed that sotagliflozin was not associated with a significant change in eGFR levels compared to the placebo at the end of the follow-up period. Finally, in the real-world setting, no difference in the occurrence of worsening renal function events between empagliflozin and insulin groups was shown.

#### 3.6.5. Hematocrit Levels

Tamaki et al. demonstrated that hemoconcentration, defined as a ≥3% absolute increase in the hematocrit levels, was significantly more frequently observed in the empagliflozin group compared to the placebo at day 7 (*p* = 0.02). Rest trials did not report a change in hematocrit levels with SGLT-2 inhibitor treatment in the acute HF setting.

#### 3.6.6. Change in Visual Analogue Scale (VAS) Dyspnoea Score

No significant change in VAS dyspnoea score with empagliflozin versus placebo was demonstrated in the EMPA-RESPONSE-AHF trial. Similarly, in the real-world setting, Perez-Belmonte and colleagues did not report any significant difference in the change in VAS dyspnoea score between empagliflozin and insulin treatment arms, despite the fact that the numeric change was greater with empagliflozin.

#### 3.6.7. Change in Quality-of-Life Indices

Empagliflozin treatment compared to placebo failed to provide a significant improvement in health status visual analog scale in the EMPAG-HF trial (*p* > 0.05). However, in the EMPULSE trial, a significant improvement in Kansas City Cardiomyopathy Questionnaire Total Symptom Score (KCCQ-TSS) was observed with empagliflozin treatment compared to the placebo. In the SOLOIST-WHF trial, trialists observed a significantly greater increase in KCCQ-12 score with sotagliflozin compared to the placebo 4 months after randomization.

#### 3.6.8. Loop Diuretics Dosage and Diuretic Efficacy

The diuretic response was significantly greater, and diuretics cumulative dosage was significantly lower with empagliflozin treatment compared to the placebo in the EMPAG-HF trial (*p* = 0.041). Similarly, in the trial performed by Charaya et al., dapagliflozin was associated with a significant decrease in average loop diuretics dosage (*p* = 0.001) compared to the placebo. The diuretic response was significantly greater both at day 15 and at day 30 with empagliflozin treatment versus placebo in the EMPULSE trial. A numeric, but not statistically significant, reduction in the cumulative dosage of loop diuretics with empagliflozin versus placebo was shown by Tamaki and colleagues (*p* = 0.071). The diuretic response did not differ at day 4 between the empagliflozin and placebo groups in the EMPA-RESPONSE-AHF trial (*p* = 0.37). Of note, even in the real-world setting, Perez-Belmonte and colleagues failed to document a significant difference in total loop diuretic dosage between the empagliflozin and insulin groups at discharge.

#### 3.6.9. Utilization Rates of Inotropes

Unfortunately, none of the studies eligible for inclusion in this comprehensive review reported the corresponding utilization rates of inotropes after initiation of treatment with SGLT-2 inhibitors compared to control in the acute HF setting.

#### 3.6.10. Usage Rates of LV Assist Devices

Similarly, none of the included studies reported any data on the need for LV assist device (LVAD) implantation after initiation of an SGLT-2 inhibitor versus control in the setting of acute HF.

A detailed summary of the above-mentioned efficacy endpoints across the included studies is provided in Table 1.

#### 3.6.11. Safety Endpoints

Since SGLT-2 inhibitors are linked with increased risk for genito-urinary tract infections and diabetic ketoacidosis (DKA), mainly euglycemic, and in this context, acute exacerbation of a chronic HF can occur, we assessed the efficacy of this drug class in the acute HF setting.

First of all, SGLT-2 inhibitors compared to control were not associated with a significantly increased risk for urinary tract infections (RR = 0.83, 95% CI; 0.53-1.30, I^2^ = 0%), as shown in Figure 6. In addition, we failed to demonstrate a significantly increased risk for DKA with SGLT-2 inhibitors versus control in the setting of acute HF (RR = 0.46, 95% CI; 0.10–2.04, I^2^ = 0%), as depicted in Figure 7. Finally, SGLT-2 inhibitor treatment did not result in a significant increase in the risk for acute kidney injury or worsening renal function, compared to control (RR = 0.87, 95% CI; 0.58–1.30, I^2^ = 42%), as demonstrated in Figure 8. Therefore, current evidence supports that the use of SGLT-2 inhibitors in the setting of acute HF is safe.

## 4. Areas for Further Research and Concluding Remarks

SGLT-2 inhibitors have provided remarkable cardiovascular benefits for subjects across the spectrum of chronic HF, regardless of the presence of underlying T2DM [67,68,69]. Patients with HFrEF seem to benefit more from SGLT-2 inhibitor treatment compared to those with HFpEF [70], and this is reflected in the relevant guidelines [4]. However, this drug class applies to a wide range of subjects with HF, with and without T2DM, and with and without concomitant obesity [71,72,73]. Recently, a vivid and ongoing discussion on the potential therapeutic role of SGLT-2 inhibitors in the setting of acute HF has started.

Indeed, SGLT-2 inhibitors are considered a novel diuretic drug class with potential applicability in acute HF [74]. As stated above, SGLT-2 inhibitors exert a number of effects that can be beneficial for patients with acute HF: they induce osmotic diuresis and natriuresis, decrease plasma volume status and interstitial fluid volume, which is expanded in HF, reduce blood pressure, and provoke hemoconcentration. In addition, they were shown to significantly improve LV function, ameliorate LV remodeling and prevent the occurrence of atrial arrhythmias, especially atrial fibrillation and atrial flutter. However, other effects need to be further explored, including improvement in arterial stiffness, myocardial flow reserve and in myocardial energetics.

In addition, despite the fact that SGLT-2 inhibitors were shown to significantly decrease pulmonary capillary wedge pressure compared to placebo among subjects with HFrEF [75], demonstrating a significant effect on central hemodynamics, none of the studies performed in the acute HF setting assessed this efficacy outcome. Another field of interest and area for further research is whether SGLT-2 inhibitors exert any effect on cardiac autonomic function, both in the acute and chronic HF setting. Cardiac autonomic function was suggested as a major tool for risk stratification of patients with cardiovascular co-morbidities [76]. Some of us have recently documented, in a meta-analysis of all relevant RCTs, that SGLT-2 inhibitor treatment among patients with T2DM does not provide any significant benefit to cardiac autonomic function [77]. Remarkably, it has not been assessed whether SGLT-2 inhibitors affect cardiac autonomic function in the acute HF setting and the direct clinical implications, thus, future trials are required to shed further light on those major pathophysiologic mechanisms that have significant prognostic implications in this patients’ population. It was also speculated that SGLT-2 inhibitors may improve renal oxygen consumption, based on their pathophysiologic mechanisms of action and that such an effect may be beneficial both in the acute and the chronic HF setting [78]; however, the only available human trial to date failed to show any significant benefit with SGLT-2 inhibitor treatment on renal oxygen consumption in subjects without underlying diabetes and hypertension [79].

Another research area of specific interest is the assessment of the effect of SGLT-2 inhibitors on circulating levels of protein biomarkers in the field of proteomics. There is increasing evidence that markers such as growth differentiation factor 15 (GDF-15) and ST2 can provide useful prognostic implications for subjects with established cardiovascular disease, including both acute and chronic HF [80,81,82]. SGLT-2 inhibitors, such as empagliflozin and dapagliflozin, were shown to significantly affect plasma levels of those protein biomarkers [83,84,85], despite the fact that in the Canagliflozin Cardiovascular Assessment Study (CANVAS) it was documented that GDF-15 reduction did not mediate the cardio-protective effects of canagliflozin [84].

However, as clearly stated in the previous section of the present comprehensive review of the relevant literature, it cannot be deduced based on current evidence that SGLT-2 inhibitors provide a significant effect on surrogate endpoints in acute HF, namely in all-cause mortality and worsening HF. In addition, there seems to be discordance between evidence retrieved from RCTs and real-world studies. Data regarding other beneficial effects, such as an increase in urine output, induction of weight loss, decrease in loop diuretics cumulative dosage, decrease in markers of congestion, such as NT-proBNP levels, improvement in subjective dyspnoea and quality of life, are encouraging; however, they are not universal across the selected studies, while their assessment at different time points does not permit pooling of raw data and synthesis of more robust evidence.

Based on their safety profile, with emphasis on renal safety, the risk for urinary tract infections and for DKA, the use of SGLT-2 inhibitors as adjunct diuretics in the setting of acute HF should not be discouraged, especially if the patients do not respond to first-line treatment options. However, current evidence seems insufficient to provide definitive answers regarding their actual role in the treatment algorithm of acute HF. Further, larger, well-designed RCTs are required to shed further light on this highly relevant topic.

## Figures and Tables

**Figure 1 life-12-02062-f001:**
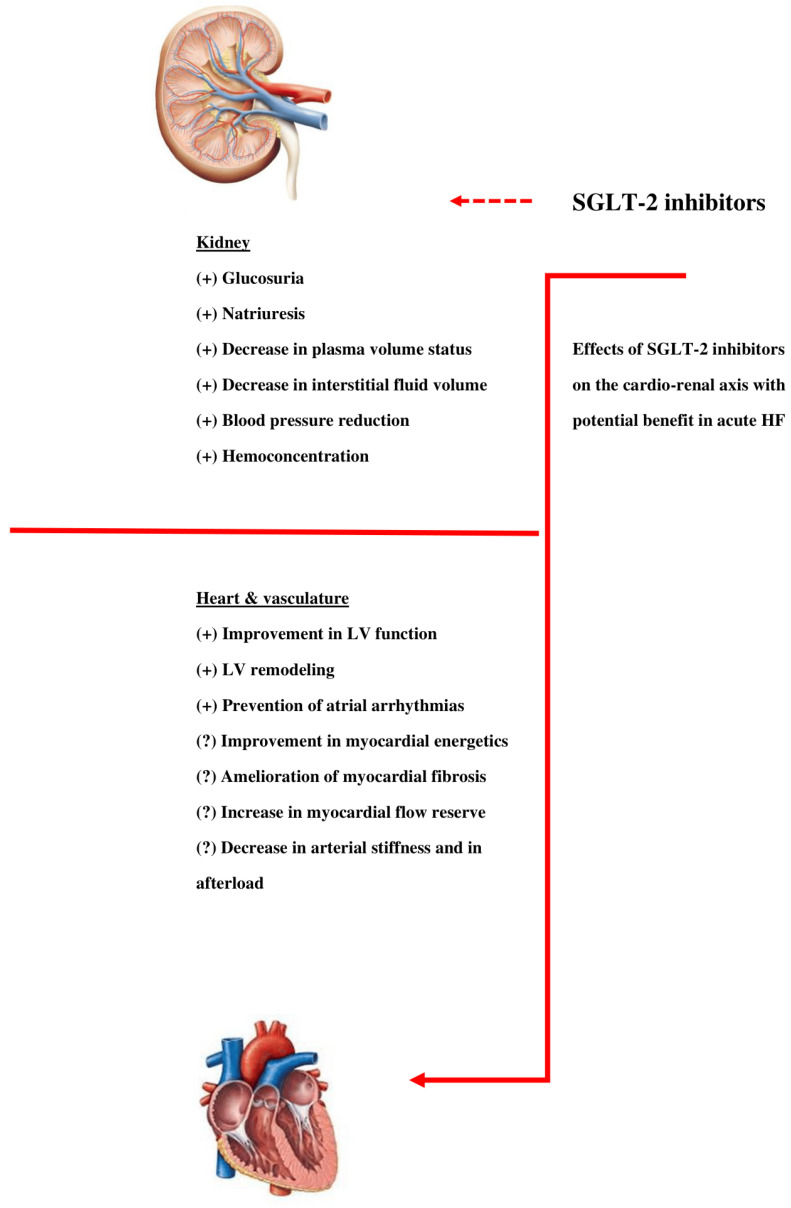
Effects of SGLT-2 inhibitors on the cardio–renal axis with potential benefit in acute HF.

**Figure 2 life-12-02062-f002:**
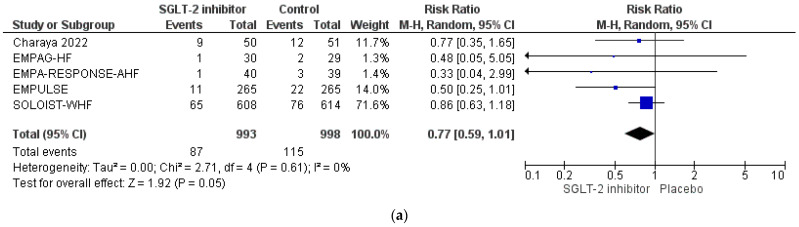
(**a**) Effect of SGLT-2 inhibitors compared to control on the risk for all-cause death: pooled analysis of eligible RCTs. (**b**) Effect of SGLT-2 inhibitors compared to control on the risk for all-cause death: pooled analysis of eligible RCTs, after exclusion of SOLOIST-WHF trial.

**Figure 3 life-12-02062-f003:**
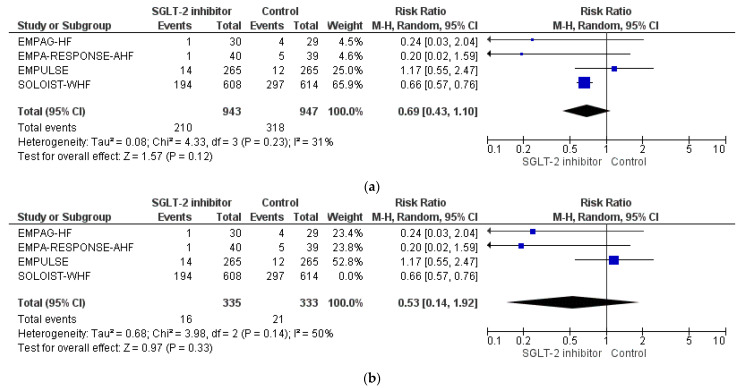
(**a**) Effect of SGLT-2 inhibitors compared to control on the risk for worsening HF: pooled analysis of eligible RCTs. (**b**) Effect of SGLT-2 inhibitors compared to control on the risk for worsening HF: pooled analysis of eligible RCTs, after exclusion of SOLOIST-WHF trial.

**Figure 4 life-12-02062-f004:**
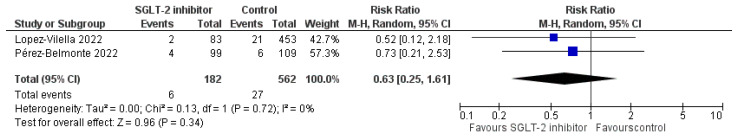
Effect of SGLT-2 inhibitors compared to control on the risk for all-cause death: pooled analysis of eligible observational studies.

**Figure 5 life-12-02062-f005:**
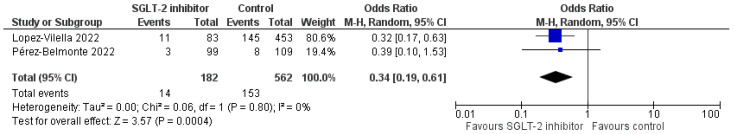
Effect of SGLT-2 inhibitors compared to control on the risk for worsening HF: pooled analysis of eligible observational studies.

**Figure 6 life-12-02062-f006:**
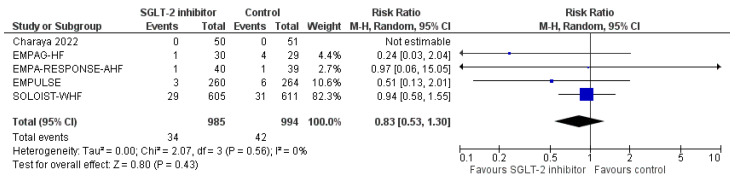
Effect of SGLT-2 inhibitors compared to control on the risk for urinary tract infections.

**Figure 7 life-12-02062-f007:**
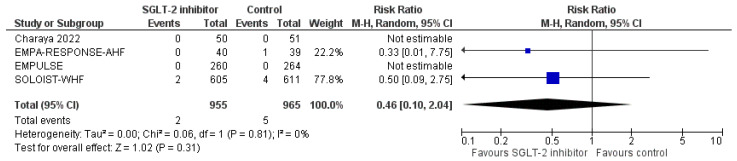
Effect of SGLT-2 inhibitors compared to control on the risk for diabetic ketoacidosis.

**Figure 8 life-12-02062-f008:**
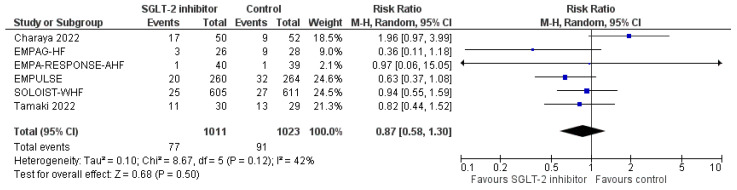
Effect of SGLT-2 inhibitors compared to control on the risk for acute kidney injury or worsening renal function.

**Table 1 life-12-02062-t001:** Summary of the effects of SGLT-2 inhibitors in the setting of acute HF.

	EMPAG-HF [52]	Charaya et al. [53]	EMPULSE [54]	Tamaki et al. [55]	EMPA-RESPONSE-AHF [56]	Perez-Belmonte et al. [58]	Lopez-Villela et al. [59]
All-cause mortality	-	-	-	Not reported	-	-	-
Worsening HF	-	Not reported	-	Not reported	-	-	+
Increase in urine output	+	Not reported	Not reported	+	+	+	Not reported
Weight loss	-	+	Not reported	-	-	Not reported	Not reported
Reduction in NT-proBNP levels	+	Not reported	+	+	-	+	Not reported
Hemoconcentration	Not reported	Not reported	Not reported	+	Not reported	Not reported	Not reported
Improvement in diuretic response/reduction in cumulative diuretic dosage	+	+	+	-	-	-	Not reported

“+”: effect reaching statistical significance. “-“: non-significant effect.

## Data Availability

Not applicable.

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
