# Peer review of "The Therapeutic Role of SGLT-2 Inhibitors in Acute Heart Failure: From Pathophysiologic Mechanisms to Clinical Evidence with Pooled Analysis of Relevant Studies across Safety and Efficacy Endpoints of Interest"

_life, 2022, doi:10.3390/life12122062_

Round 1

Reviewer 1 Report

The present review is aimed at tackling an important aspect of the novel use of SGLT2i in HF, namely the setting of acute HF. In this prospective the review is comprehensive and presents properly the currently knowledge on the topic.

Overall, I have few comments only.

In the setting of AHF pressure more than volume is the matter. Both in HFpEF and rEF end diastolic pressure may significantly increase for small volume changes that may not be detectable on body weight. Despite some contrasting real life data the option remains that pulmonary pressure might be the parameter to monitor to tackle acute WHF events. In this prospective I’m wondering if any data can be extracted from the available literature.

Second is the aspect concerning autonomic responses to chronic and acute HF. In many HF database not even HR is collected while the key therapy is based on neurohumoral interventions. In the acute setting HR is almost always elevated but effective afterload reduction by interventions not affecting renal function might result in rapid amelioration of the cardiorenal reflexes that might be detected by looking to HR, or HRV that might be extracted from ECG monitoring. I honestly do not aspect to see any data on this but it might be appropriate to use conclusive remarks also to highlight what current knowledge is missing.

Third is the hypothesis that SGLTi may significantly decrease renal oxygen consumption. Such an action in acute HF would certainly contribute to a better and more rapid outcome.

Last but not least is the detrimental use of inotropic agents. Any data about lesser use of these agents (Class III in guidelines) in the AHF?

As general comment I would suggest to edit the introductory part of the review. Too many short paragraphs do not help. Please try to compile all these concepts in a single introductory section.

Author Response

The present review is aimed at tackling an important aspect of the novel use of SGLT2i in HF, namely the setting of acute HF. In this prospective the review is comprehensive and presents properly the currently knowledge on the topic.

Overall, I have few comments only.

In the setting of AHF pressure more than volume is the matter. Both in HFpEF and rEF end diastolic pressure may significantly increase for small volume changes that may not be detectable on body weight. Despite some contrasting real life data the option remains that pulmonary pressure might be the parameter to monitor to tackle acute WHF events. In this prospective I’m wondering if any data can be extracted from the available literature.

Answer: We would like to heartily thank the reviewer for this important comment. Indeed, only one trial has assessed the effect of SGLT-2 inhibitors on pulmonary capillary wedge pressure (PCWP) after right-heart catheterization in patients with HFrEF (Omar et al., Circulation 2021), while this endpoint has not been assessed in any of the available studies performed in the acute HF setting. A relevant comment has been added in the “Concluding remarks” section.

Second is the aspect concerning autonomic responses to chronic and acute HF. In many HF database not even HR is collected while the key therapy is based on neurohumoral interventions. In the acute setting HR is almost always elevated but effective afterload reduction by interventions not affecting renal function might result in rapid amelioration of the cardiorenal reflexes that might be detected by looking to HR, or HRV that might be extracted from ECG monitoring. I honestly do not aspect to see any data on this but it might be appropriate to use conclusive remarks also to highlight what current knowledge is missing.

Answer: We thank the reviewer for this significant and targeted comment. Indeed, none of the eligible studies for inclusion in this review has assessed the effects of SGLT-2 inhibitors on HRV in the acute HF setting, and thus, this represents an area for further research. A comment has been added in the final section of the manuscript.

Third is the hypothesis that SGLTi may significantly decrease renal oxygen consumption. Such an action in acute HF would certainly contribute to a better and more rapid outcome.

Answer: We cordially thank the reviewer for this important comment. We now comment on this interesting aspect within the final section of our revised manuscript.

Last but not least is the detrimental use of inotropic agents. Any data about lesser use of these agents (Class III in guidelines) in the AHF?

Answer: We thank the reviewer for this comment. Unfortunately, none of the eligible studies included in this review reported the usage rates of inotropic agents after initiation of SGLT-2 inhibitor treatment compared to control.

As general comment I would suggest to edit the introductory part of the review. Too many short paragraphs do not help. Please try to compile all these concepts in a single introductory section.

Answer: We cordially thank the reviewer for this comment. We have attempted to shorten at some extent the introductory section and to compile paragraphs together.

Reviewer 2 Report

In this review, Dimitrios Patoulias et al. approached a very actual topic: the role of SGLT2 in heart failure (HF). Given that the majority of previous literature data referred strictly to their use in chronic HF, I consider that the authors have converged to review an interesting yet challenging subject: the use of SGLT2 inhibitors in acute heart failure. The manuscript targeted the main pathophysiological pathways influenced by those molecules, however there are certain more points to be focused on:

1.      A subtopic concerning the adverse effects and the subsequent mechanisms of SGLT2i is highly required. It is well-known that (urinary) infections represent a common cause for HF decompensation. In an acute setting, each aspect should be carrefully considered, and possibly corrected.  

2.      The authors stated that SGLT2 inhibitors were associated with an improvement of LV ejection fraction and the reduction of LV mass. Please provide additional references concerning the involved mechanisms.

3.      Further discussions are required concerning the biomarkers dynamics in acute heart failure. The authors provided references strictly concerning NT-proBNP. Given the novel promising molecules, such as ST2 or GDF-15, I suggest to include them in the biomarkers section influenced (or not) by SGLT2 i. Example of how ST2 acts in heart failure: Miftode RS, Petriș AO, Onofrei Aursulesei V, Cianga C, Costache II, Mitu O, Miftode IL, Șerban IL. The Novel Perspectives Opened by ST2 in the Pandemic: A Review of Its Role in the Diagnosis and Prognosis of Patients with Heart Failure and COVID-19. Diagnostics (Basel). 2021 Jan 26;11(2):175. doi: 10.3390/diagnostics11020175. 

4.      A certain improvement would be provided by a new section concerning the correlation between the administration of SGLT2 inhibitors in acute settings and the need for left ventricular assist devices.

5.      The manuscript should be reviewed by a native English speaker. Some phrases and constructions are odd and “non-natural”.

Best regards,

The Reviewer

Author Response

In this review, Dimitrios Patoulias et al. approached a very actual topic: the role of SGLT2 in heart failure (HF). Given that the majority of previous literature data referred strictly to their use in chronic HF, I consider that the authors have converged to review an interesting yet challenging subject: the use of SGLT2 inhibitors in acute heart failure. The manuscript targeted the main pathophysiological pathways influenced by those molecules, however there are certain more points to be focused on:

  1. A subtopic concerning the adverse effects and the subsequent mechanisms of SGLT2i is highly required. It is well-known that (urinary) infections represent a common cause for HF decompensation. In an acute setting, each aspect should be carrefully considered, and possibly corrected.  

Answer: We heartily thank the reviewer for this very important and targeted comment, aiming at the substantial improvement of our manuscript’s overall quality. We have added safety analyses in our revised manuscript, assessing the effect of SGLT-2 inhibitors versus control on the risk for major safety endpoints in the setting of acute HF. Thank you once again for this crucial comment.

  1. The authors stated that SGLT2 inhibitors were associated with an improvement of LV ejection fraction and the reduction of LV mass. Please provide additional references concerning the involved mechanisms.

Answer: We thank the reviewer for this important comment. A short comment in the corresponding section of the revised manuscript has now been added, citing relevant references of significance.

  1. Further discussions are required concerning the biomarkers dynamics in acute heart failure. The authors provided references strictly concerning NT-proBNP. Given the novel promising molecules, such as ST2 or GDF-15, I suggest to include them in the biomarkers section influenced (or not) by SGLT2 i. Example of how ST2 acts in heart failure: Miftode RS, Petriș AO, Onofrei Aursulesei V, Cianga C, Costache II, Mitu O, Miftode IL, Șerban IL. The Novel Perspectives Opened by ST2 in the Pandemic: A Review of Its Role in the Diagnosis and Prognosis of Patients with Heart Failure and COVID-19. Diagnostics (Basel). 2021 Jan 26;11(2):175. doi: 10.3390/diagnostics11020175.

  1. A certain improvement would be provided by a new section concerning the correlation between the administration of SGLT2 inhibitors in acute settings and the need for left ventricular assist devices.

Answer: We cordially thank the reviewer for this targeted and very important comment. Unfortunately, none of the eligible studies reported any data on the need for LVAD implantation in the setting of acute HF after initiation of a SGLT-2 inhibitor versus control. A relevant section has been added in the revised manuscript.

  1. The manuscript should be reviewed by a native English speaker. Some phrases and constructions are odd and “non-natural”.

Answer: Thank you for this comment. Manuscript has been extensively reviewed for English language errors and revised accordingly.

Best regards,

The Reviewer

Reviewer 3 Report

            The authors investigate the role of SGLT2i on acute heart failure (AHF). First, the authors present a comprehensive review of the mechanisms of action of the benefits of SGLT2i on acute HF (and not only on chronic HF). Then, the authors present meta-analysis of the few trials of SGLT2i on AHF.

            SGLT2i have exhaustively been studied on chronic HF, but acute HF has remained mostly unexplored (except for a few exception). Therefore the systematic summary of these few studies on acute HF is of high novelty and interest. Ofnote, the manuscript is of high clinical relevance. The manuscript is pretty comprehensive (only minor additions are required), is balanced, and it reads well.

            Major issues:

-          Arterial stiffness: The authors report no change in arterial stiffness in patients at high CV risk. However, their review is about heart failure. Therefore, they should mention that empagliflozin reduces arterial stiffness in patients with heart failure (please quote JACC Heart Fail. 2021 Aug;9(8):578-589).

-          Energetics: The authors should mention that

o   In a relevant porcine model of HF, empagliflozin has demonstrated to switch myocardial fuel utilization away from glucose towards the consumption of free fatty acids, ketone bodies and branched-chain amino acids, which improve cardiac energetics (please quote J Am Coll Cardiol. 2019 Apr 23;73(15):1931-1944).

o   In human HFrEF patients, a proteomic subanalysis of the DEFINE-HF trial supports these findings of increased free fatty acid and ketone bodies consumption (please quote Circulation 2022 Sep 13;146(11):819-821.).

-          In the mechanism section, the author should explain that SGLT2i improve diastolic function which results in a reduction in filling pressures (please quote JACC Cardiovasc Imaging. 2021 Feb;14(2):393-407).

Author Response

       The authors investigate the role of SGLT2i on acute heart failure (AHF). First, the authors present a comprehensive review of the mechanisms of action of the benefits of SGLT2i on acute HF (and not only on chronic HF). Then, the authors present meta-analysis of the few trials of SGLT2i on AHF.

            SGLT2i have exhaustively been studied on chronic HF, but acute HF has remained mostly unexplored (except for a few exception). Therefore the systematic summary of these few studies on acute HF is of high novelty and interest. Ofnote, the manuscript is of high clinical relevance. The manuscript is pretty comprehensive (only minor additions are required), is balanced, and it reads well.

            Major issues:

-          Arterial stiffness: The authors report no change in arterial stiffness in patients at high CV risk. However, their review is about heart failure. Therefore, they should mention that empagliflozin reduces arterial stiffness in patients with heart failure (please quote JACC Heart Fail. 2021 Aug;9(8):578-589).

Answer: We cordially thank the reviewer for this important comment, which has been addressed in the corresponding section of the revised manuscript.

-          Energetics: The authors should mention that

o   In a relevant porcine model of HF, empagliflozin has demonstrated to switch myocardial fuel utilization away from glucose towards the consumption of free fatty acids, ketone bodies and branched-chain amino acids, which improve cardiac energetics (please quote J Am Coll Cardiol. 2019 Apr 23;73(15):1931-1944).

o   In human HFrEF patients, a proteomic subanalysis of the DEFINE-HF trial supports these findings of increased free fatty acid and ketone bodies consumption (please quote Circulation 2022 Sep 13;146(11):819-821.).

Answer: We cordially thank the reviewer for those two relevant comments. We have now revised the section “Myocardial energetics” accordingly, citing the provided experimental and human studies.

-          In the mechanism section, the author should explain that SGLT2i improve diastolic function which results in a reduction in filling pressures (please quote JACC Cardiovasc Imaging. 2021 Feb;14(2):393-407

)

Answer: We would like to thank the reviewer for this major comment. A short comment has been added in the corresponding section of the manuscript.